# Preparation and Photocathodic Protection Properties of ZnO/TiO_2_ Heterojunction Film Under Simulated Solar Light

**DOI:** 10.3390/ma12233856

**Published:** 2019-11-22

**Authors:** Xiong Zhang, Guanghui Chen, Weihua Li, Dianwu Wu

**Affiliations:** 1School of Materials Science and Engineering, Tongji University, No. 4800 Caoan Road, Shanghai 201804, China; zhangxiong@tongji.edu.cn; 2College of chemical engineering, Qingdao University of science & technology, No.53 Zhengzhou Road, Qingdao 266042, China; guanghui@quest.edu.cn; 3School of Chemical engineering and technology, Sun Yat-Sen University Zhuhai Campus, tangjiawan, Zhuhai 519082, China; liweihua3@mail.sysu.edu.cn

**Keywords:** ZnO, TiO_2_, heterojunction, photocathodic protection, corrosion protection

## Abstract

In this work, a novel double layer made of ZnO nanorod arrays and TiO_2_ nanorod arrays with anticorrosion function were successfully prepared on fluorine-doped tin oxide (FTO) substrate by a simple low-temperature solvothermal method. As compared with the pure TiO_2_ and pure ZnO film, the combination of the two films presented higher photocathodic protection performance for 316 stainless steel (316 SS) and Q235 carbon steel in 3.5 wt% NaCl solution. The composite film with ZnO nanoparticles layer between ZnO nanorod arrays and TiO_2_ nanorod arrays exhibited the best photocathodic performance, which lowered the open circuit potential (OCP) of 316 SS and Q235 to −991 mV, −1066 mV, respectively. The results demonstrated that the formation of the uniform heterojunction film and the small difference in band alignment played important roles in the promotion of photocathodic protection performance.

## 1. Introduction

Metal corrosion is a spontaneous process which causes great losses yearly, including economic losses, environmental damages, losses of life and injury, loss of efficiency, etc. Several corrosion control strategies have been applied, and impressed current cathodic protection, sacrificial anode and the coating are the three most widely used and most effective methods. However, impressed current cathodic protection demands a stable current supply, which is not suitable for use in remote places, such as offshore areas. The materials used in sacrificial anodes are expensive and consume a great amount of resources every year. The coating cannot tolerate a few breakages in the coating, because this accelerates the rate of corrosion at the site of breakage. Therefore, a sustainable and environmentally friendly technology for corrosion control was placed on the agenda.

In 1994, Fujisawa [1] originally found the photocathodic protection effect in TiO_2_ film. He found the emergence of electron-hole pairs in titanium dioxide (TiO_2_) film when irradiated with ultraviolet (UV) light, and the OCP of the coupled metal shifted to the negative potential, which lowered and even prevented the corrosion of the coupled metal. Photocathodic protection denotes that the electrons in photoanodes (*n*-type semiconductors in general) are excited by solar light and migrate to the coupled metal in a direct or indirect manner, resulting in the cathodic polarization of metal. Furthermore, the anodic dissolution reaction of the metal is inhibited, and the material is protected by cathodic current. The biggest advantage of photocathodic protection technology is the use of clean and inexhaustible solar energy for corrosion protection. The main applications include steel structures (stainless steel [2,3,4,5,6,7] and carbon steel [8,9]), copper structures [10], and other possible metals in offshore areas and other marine environments, where there is no reliable power supply for cathodic protection. During the process, the photoanode material that acts as the electronic transmission carrier is not consumed (non-sacrificed photoanode) from beginning to end, and a semiconductor with good stability can make the system work economically. To maintain the sustainability of the protection, an adequate and continuous supply of electrons is essential. The photoanode materials coupled with another semiconductor (WO_3_ [11], SnO_2_ [5,12], V_2_O_5_ [13], MoO_3_ [14]) which has electron storage properties to make it sustainable under dark conditions have been proved to be effective methods. Additionally, preparing the photoanode film directly on the surface of the metal might be another photocathodic protection model, the film not only serving as the photoanode, protecting metals in daylight, but also serves as a coating that can insulate the metal from the corrosion environment. Another advantage is that a coating with photocathodic protection properties can allow for the existence of coating damage, where metal corrosion would accelerate in traditional coating due to the negative potential of the total metal under irradiation. Photoanode films with dual function can protect metal from corrosion effectively, and it is possible to construct a long-life, near zero energy consumption, and maintenance-free corrosion protection system.

Photocathodic protection technology is green and sustainable, and is one of the most promising anticorrosion technologies; in recent years, research into the photocathodic protection effect using several semiconductors has attracted widespread attention globally. Among these, TiO_2_ [2,3,4,5,6,7] has widely attracted the interest of researchers due to its good stability, lack of toxicity, abundances of resources, and inexpensiveness. However, electrons in TiO_2_ can only be excited by UV light due to its wide band gap (~3.2 eV). In addition, electron–hole pairs in TiO_2_ have a relatively high recombination rate and a low charge mobility. Therefore, in order to improve the photo-induced charge separation, promote the electron yield, and thus improve the photocathodic protection performance, it is necessary to make some modifications to TiO_2_, and the combination of TiO_2_ with various materials has been studied in recent years. For example, its combination with oxide semiconductors (Fe_2_O_3_ [8], NiO [15], β-Bi_2_O_3_ [16]), sulfides (Bi_2_S_3_ [2], SnS [17], CdS [18], Ag_2_S [19], MnS [20]), selenides (CdSe [4], Bi_2_Se_3_ [21], NiSe_2_ [22], WSe_2_ [23]), perovskites (SrTiO_3_ [3]), and other materials (CoFe_2_O_4_ [24], ZnFeO_4_ [7], ZnIn_2_S_4_ [9], ZnWO_4_ [25], BiFeO_3_ [26]). Other heterojunction materials used in the PEC field have also been deeply studied to improve their photo-electric conversion efficiency and stability, including for water splitting [27,28] and solar redox flow battery applications [29,30,31], which is can be used as a reference in photocathodic protection systems. Due to Zinc oxide (ZnO) and TiO_2_ having the same wide band gap (~3.2 eV), and due to the small difference in the position of the conduction band (*CB*) and valence band (*VB*) In ZnO compared with TiO_2_, the migration of electrons and holes might be relatively easier between TiO_2_ and ZnO. Additionally, ZnO has high electron mobility, which is two orders of magnitude higher than TiO_2_ [32,33], ZnO combined with TiO_2_ might be an appropriate photoanode choice in a photocathodic protection system.

In this paper, we successfully prepared a ZnO/TiO_2_ heterojunction double layer film by adopting a simple and low-cost solvothermal method for ZnO nanorod arrays growing on TiO_2_ nanorod arrays. The ZnO nanoparticle layer between ZnO and TiO_2_ is the key to forming a uniform photoanode film, and thus promoting photocathodic protection performance. The small difference in band alignment between ZnO and TiO_2_ made a sharp separation between photogenerated charge carriers, thus effectively providing photogenerated cathodic protection for 316 SS and Q235.

## 2. Experimental

All of the chemical reagents used in this study were analytical reagents and without further purification. All of the aqueous solutions were prepared using double distilled water.

### 2.1. Preparation of TiO_2_ Nanorod Arrays

TiO_2_ nanorod arrays were prepared using a solvothermal method reported previously [34], with minor modification. Firstly, FTO (≤15 Ω/sq, Zhuhai Kaivo Optoelectronic Co., Ltd., Zhuhai, China) substrates were ultrasonic cleaned with deionized water (DI water), acetone, ethanol, and DI water for 30 min, respectively. Then they were dried in the vacuum drying oven at 80 °C for 30 min. Secondly, 12 mL of concentrated hydrochloric acid (HCl, 38.5%) was added to 12 mL of DI water with magnetic stirring for 10 min, and 400 µL tetrabutyl titanate (TIA, C_16_H_36_O_4_Ti) was added to the solution above with magnetic stirring for 20 min. Then, the mixed solution was transferred to a Teflon-lined autoclave (50 mL). The cleaned F TO glass was inserted into the Teflon-lined autoclave with the conducting side facing obliquely downwards, and then the autoclave was placed in the drying oven. After reacting at 150 °C for 18 h, the substrate was washed with DI water several times, dried in air, and annealed at 450 °C for 2.0 h. Finally, milk-white TiO_2_ film was produced on the FTO substrate.

### 2.2. Preparation of ZnO/TiO_2_ Heterojunction Double Layer Film

The ZnO/TiO_2_ double layer film was prepared by a solvothermal method (depicted in Figure 1a). The first step was to prepare the ZnO seed layer (ZnO nanoparticle film) on TiO_2_/FTO substrate by spin coating. The zinc nitrate hexahydrate (Zn(NO_3_)_2_·6H_2_O, 0.01 M) and hexamethylenetetramine (C_6_H_12_N_4_, HMT, 0.01M) aqueous solution were used as the precursor. The spin parameter was set to the third gear, and the velocities were set to 500 r/min, 2000 r/min, and 5000 r/min, respectively; the rotational times for each gear were all set at 10 s. After that, the substrate was sintered in a muffle furnace at 450 °C for 10 min. This process was repeated three times in order to form a uniform ZnO nanoparticle layer on TiO_2_ nanorod arrays. Secondly, ZnO nanorod arrays were grown vertically from the nanoparticle seeds by immersing ZnO seed-coated TiO_2_/FTO substrate in an aqueous solution of 0.025 M Zn(NO_3_)_2_·6H_2_O and 0.025 M HMT. The reaction temperature and time were set to 80 °C and 12 h in the water bath, and then thorough rinsing was carried out with DI water several times to eliminate the residual salts, followed by drying at 80 °C for 1.0 h. The as-prepared sample was labeled as ZT2 film. For comparison, ZnO nanorod arrays grown on TiO_2_/FTO substrate without ZnO seed layer were labeled as ZT1, and ZnO nanorod arrays grown on ZnO seed-coated FTO substrate were labeled as ZnO.

### 2.3. Characterization 

Morphological characterization was performed by scanning electron microscopy (SEM, ZEISS EV018, Oberkochen, Germany). The crystalline structures of the samples were recorded by X-ray diffraction (XRD, Bruker AXS D8 ADVANCE, Madison, USA) with a copper X-ray source (Cu–Kα, 50 kV, 250 mA). The optical properties were determined using a fluorospectrophotometer (Hitachi F-4600, Tokyo, Japan), UV-Vis-NIR spectrophotometer (Agilent Cary 5000, Palo Alto, USA). The OCP and photocurrent density of the photoanodes were determined using an electrochemical workstation (Gamry Reference 3000, Philadelphia, USA).

### 2.4. Electrode Fabrication and Photoelectrochemical Measurements 

Metal electrode: Here, we chose the widely used steel (316 SS and Q235) to be the object of our research. First, the metal cubes (10 mm × 10 mm × 10 mm, with element contents as shown in Table 1) were polished with sand paper of 600, 1200 and 2000 mesh in sequence using a metallographic polishing machine. Second, to ensure good contact, the metal was connected with copper wire by tin welding. Third, keeping one side of the cube metal (to be studied, 10 mm × 10 mm) exposed, the other sides were encapsulated, and the joint was welded with ethoxyline resin (ER) in case of corrosion. Before each examination, the exposed side of the iron cube was polished with sand paper (2000 mesh) to ensure the consistency and veracity of the results.

Photoanode: The total size of the photoanode was 10 mm × 13 mm (in order to retain a surface ratio of 1:1 between photoanode and metal electrode; the size of the photoanode film was 10 mm × 10 mm, and 10 mm × 3 mm blank FTO was preserved with no photoanode film). Then the conductive side of the blank FTO was connected with the copper wire by silver conductive adhesive to ensure good contact. Finally, the entire region of the blank FTO was encapsulated with ER.

Photoelectrochemical measurements: To observe the potential change in the metal electrode in order to evaluate the photocathodic protection effect of the prepared films, we designed an experimental setup as shown in Figure 1b. The experimental setup is a double electrolytic cell system composed of a photoelectric cell and a corrosion cell. The photoelectrode and metal electrode were placed into the photoelectric cell and corrosion cell, respectively. The electrolyte in the photoelectric cell is Na_2_S (0.1 M) and Na_2_SO_3_ (0.1 M). Because ZnO is quite a PH-sensitive material, it can only maintain stability in neutral and weak alkaline environments. Due to the fact that the applications mainly focus on offshore areas and other marine environments, the electrolyte in the corrosion cell was a simulated seawater solution (3.5 wt% NaCl). The measurement of OCP and photocurrent density was conducted by Zero Resistance Ammeter (ZRA Mode) on a Gamry Reference 3000. The measurements were carried out under a tunable simulated solar light source with a 300 W Xenon lamp (Microsolar 300, Beijing Bofeilai Co., Beijing, China). The distance between the lamp and the photoanode was 200 mm, and the power we used was 50 W, with a corresponding intensity of 50 mW/cm^2^. Mott-Schottky and Bode-phase plots of the photoanodes were performed for a conventional three-electrode system, in which SCE served as RE, platinum as CE, the photoanode as WE, and 0.1 M Na_2_SO_4_ as the electrolyte. All electrochemical measurements were performed at ambient temperature.

## 3. Results and Discussion

### 3.1. Morphology and Crystal Structure Analysis 

The surface morphologies of TiO_2_ nanorod arrays, ZnO nanorod arrays, and two types of ZnO/TiO_2_ heterojunction films are depicted in Figure 2. It was shown that the TiO_2_ nanorod grew vertically on FTO substrate with a thin tip and a thick root. The diameter of the tip was 70–80 nm (Figure 2a), the diameter of the root and the length of the rod were approximately 300 nm and 4 µm, respectively (Figure 2b). ZnO nanorods appeared to be comparatively regular hexagonal prisms, and grew perpendicularly on the FTO substrate like TiO_2_ nanorod arrays. The diameter of the ZnO nanorod was about 80 nm (Figure 2c), the thickness of the ZnO nanoparticle layer and the length of the ZnO rod were about 300 nm and 1.5 µm, respectively (Figure 2d). ZnO/TiO_2_ heterojunction arrays without a ZnO nanoparticle layer are shown in Figure 2e; the ZnO rod grew in a disorderly fashion on TiO_2_ nanorod arrays like a weed infestation, and the diameters of ZnO rods were nonuniform, ranging from 70 nm to 700 nm. Additionally, we can clearly observe some small pits labeled with yellow square frames in Figure 2e, and the nanorods in the bottom of these vacancies should be TiO_2_. Due to the difference between TiO_2_ and ZnO, the nucleation of ZnO in the surface of TiO_2_ crystal will not homogeneous, so the growth of ZnO nanorod arrays grown on TiO_2_ might not be homogeneous. After being coated with the ZnO seed layer, ZnO nanoparticles will be a site at which the nucleation of ZnO nanorods is possible, which is conducive for the homogeneous growth of ZnO nanorod arrays. Figure 2f shows that the ZnO nanorods in ZnO/TiO_2_ heterojunction arrays with ZnO nanoparticle layer were relatively uniform, with an average diameter of about 90 nm.

Figure 3 presents the XRD patterns of different photoanode films. The diffraction peaks at 25.28°, 37.77°, 48.02°, 53.80°, 55.22 and 62.85° were in accordance with the anatase TiO_2_ (JCPDS card No.21-1272), corresponding to the (101), (004), (200), (105), (211) and (204) crystal faces of TiO_2_, respectively. The diffraction peaks at 31.78°, 34.44°, 36.26°, 47.54°, 56.58°, 62.85°, 66.35 and 67.93° accorded with the (100), (002), (101), (102), (110), (103), (200), and (112) crystal faces of ZnO (JCPDS card No. 36-1451), respectively. The diffraction peaks of the composite films (ZT1 and ZT2) all appeared in the patterns, proving the formation of the heterojunction film between TiO_2_ and ZnO. The diffraction peaks of ZT2 are sharper compared with ZT1, which can be attributed to the homogeneous growth of ZT2 compared with ZT1.

### 3.2. Light Absorption and Charge Mobility

Figure 4a shows the UV-Vis absorptance spectra of different photoanode films. Owing to the similar band gap, these photoanode films showed similar absorption spectra in the measured wavelength range. The absorption band edges of TiO_2_ and ZnO were at approximately 380 nm and 395 nm, respectively, and the absorption of the composite film (ZT1 and ZT2) between ZnO and TiO_2_ were slightly red-shifted. Compared to the relatively lower absorption intensities in the ultraviolet range, the ZT1 film and ZT2 film had higher light absorption intensities in the visible light range (more than 400 nm), which means that the ability to utilize visible light was promoted due to the heterojunction of ZnO and TiO_2_. The band gap of the photoanode can be roughly estimated suing the following equation for a semiconductor: *αhν* = A (*hν* – E_g_)^η^ [35]; where *α, h, ν,* A, E_g_ and η represent the optical adsorption coefficient, the Planck constant, the frequency of light, a constant, the band gap of the semiconductor, and a characteristic of the type of electron transition process (η = 1/2 for a direct semiconductor, η = 2 for an indirect semiconductor), respectively. Figure 4b shows the plots of (*αhν*)^1/2^ vs. photoenergy. The estimated band gap values of TiO_2_, ZnO, ZT1 and ZT2 films were approximately 3.17 eV, 3.10 eV, 2.99 eV and 2.93 eV, respectively. It is evident that the band gap of ZT1 film and ZT2 film had narrowed, and that these two films could absorb light of longer wavelengths.

Figure 5 presents the PL spectra of four different films. The PL spectra consist of two emission peaks: one is a UV emission peak centered at about 390–395 nm, which belongs to the exciton recombination corresponding to the near-band-edge emission of the ZnO; and the other one is a distinct peak centered at 465 nm that originates from oxygen vacancies or defects in samples [36]. Compared with TiO_2_, the UV-emission peaks were slightly blue-shifted, which was in agreement with the result of UV-Abs. PL intensity is typically used to evaluate the photogenerated electron-hole recombination, and the higher the PL intensity, the higher the electron-hole recombination efficiency [37,38]. As shown in Figure 5, the PL intensities of ZnO and TiO_2_ film are comparatively high. However, the PL intensities of the composite films were all decreased, and ZT2 film had the lowest PL intensity, implying that the excited electron-hole recombination had been suppressed, and that the recombination efficiency in ZT2 film was the lowest.

The frequency value of the characteristic peak in the Bode-phase plots is typically used to evaluate the lifetime of the photogenerated electrons, according to the empirical equation: τ ≈ 1/(2π*f*_max_) [39,40], where τ is the effective electron lifetime, *f*_max_ is the frequency value of the characteristic peak in the Bode-phase plots, and τ is inversely proportional to *f*_max_. From Figure 6, the *f*_max_ of the TiO_2_, ZnO, ZT1, ZT2 films was 9.93 Hz, 5.01 Hz, 3.16 Hz, and 1.58 Hz, respectively. The lifetime of the TiO_2_, ZnO, ZT1, ZT2 films was 0.016, 0.032, 0.05, and 0.10 s, respectively. Therefore, the lifetime of the photogenerated electrons in the ZT2 composite film was the longest (reaching 6 times that of the TiO_2_ film), which is a result of the inhibited charge carrier recombination and enhanced charge separation efficiency in the composite film.

### 3.3. Photocathodic Protection Performances 

To validate the photocathodic protection properties of the as-prepared samples, the OCP and photocurrent density were measured of the galvanic couple between 316 SS electrode and different films under intermittent simulated sunlight in 3.5 wt% NaCl solution. According to the principle of conventional cathodic protection for metal protection, when galvanic current is impressed into the metal, the surface of the metal will change entirely into the cathode, the potential of the coupled metal will polarize and the original potential will deviate to the negative position, lowing the metal corrosion potential to the protective potential, and ultimately suppress the metal corrosion. As shown in Figure 7a, all samples have an infinitesimal (near zero) photocurrent under dark conditions. The photocurrent densities all increased rapidly due to the photoelectric effect when the light was switched on, indicating that the photogenerated electrons flow from the photoanodes to the metal electrode. The order of the photogenerated current of the coupled photoanodes is as follows: ZT2 (151 µA/cm^2^) > ZT1 (108 µA/cm^2^) > TiO_2_ (52 µA/cm^2^) > ZnO (26 µA/cm^2^). After switching off the light, the photocurrent became very small, and when the light was switched on again, the photocurrent rapidly increased. However, to a slightly lower degree, in relative terms, than when first switching on. After four iterations, the values of the photogenerated current were as follows: ZT2 (126 µA/cm^2^) > ZT1 (81 µA/cm^2^) > TiO_2_ (33 µA/cm^2^) > ZnO (24 µA/cm^2^). Compared with ZnO and TiO_2_, the photocurrent of ZT1 film and ZT2 film all increased, which might be due to the heterojunction structure composed of ZnO and TiO_2_, thus reducing the recombination probability of the photogenerated electron-holes. Larger photogenerated photocurrent density might mean indicate the better separation efficiency of the photogenerated electron-holes, and ZT2 film might be the optimal photoanode for providing protection to 316 SS.

The values of OCP express the thermodynamics trend of metal corrosion: the more negative the potential, the smaller the trend of metal suffering corrosion. The corrosion potential (E_corr_) of 316 SS in 3.5 wt% NaCl solution is approximately −180 mV, and the OCPs decrease to some degree when coupled with different photoanodes. The OCPs shifted to the negative potential immediately once the light was switched on, which was caused by the impressed current generated by the photoelectrons; this phenomenon verified the cathodic polarization of the metal. After 500 s’s irradiation, the OCP values became relatively stable, which may be attributed to the balance between the generation and recombination of the photogenerated electron-holes [41]. According to polarization theory, the larger the impressed current is, the stronger the cathodic polarization and the more negative potential the metal has. As a result of the values of the photogenerated current of the different coupled photoanodes, the ZT2 film should have the greatest negative potential. From Figure 7b, the order of the relative equilibrium potentials of the photoanodes was: ZT2 (−991 mV vs SCE) < ZT1 (−863 mV vs SCE) < TiO_2_ (−787 mV vs SCE) < ZnO (−486 mV vs SCE). The OCP shifted to the original position when the light was switched off, but was still more negative than the E_corr_ of 316 SS. We can conclude that all of the photoanodes are able to provide effectively protection for 316 SS, while the ZT2 film is the best scheme.

We also studied the photocathodic protection properties of Q235 using different photoanodes. Figure 8 shows the photocurrent density and OCP of the galvanic couple between Q235 electrode and different films under intermittent simulated sunlight in 3.5% NaCl solution. Similar to 316 SS, the photocurrent density of the coupled photoanodes increased when the light was switched on; Figure 8a shows that the photogenerated current density of the coupled photoanodes was: ZT2 (238 µA/cm^2^) > ZT1 (111 µA/cm^2^) > TiO_2_ (62 µA/cm^2^) > ZnO (17 µA/cm^2^). The E_corr_ of Q235 in 3.5% NaCl solution was approximately −630 mV, so we simply draw a straight line. When coupling with different photoanodes, the OCPs of all films were initially higher than the E_corr_ of Q235, indicating that none of the films could initially provide sufficient cathodic protection for Q235 under dark conditions. When the light was switched on, the OCP shifted sharply to the negative position, which was lower than the E_corr_ of Q235 for all films, and the photogenerated current successfully gave rise to the polarization of Q235. From Figure 8b, the order of the OCPs of the photoanodes was: ZT2 (−1066 mV vs SCE) < ZT1 (−1022 mV vs SCE) < TiO_2_ (−893 mV vs SCE) < ZnO (−832 mV vs SCE). We can therefore conclude that all of the photoanodes could provide effective protection for Q235 under irradiation; ZT1 and ZT2 could provide continuously protection under dark conditions, and ZT2 film had the best protective effect.

### 3.4. Possible Photocathodic Protection Mechanism 

Figure 9a shows the Mott-Schottky plots of the photoanode films in 0.1 M Na_2_SO_4_ solution in dark state. The Mott-Schottky plot has typically been used to indicate the *p* or *n* characteristic of a semiconductor and calculate the corresponding conduction band. The positive slopes of the Mott-Schottky plots present *n*-type semiconductor characteristic of all the samples. The flat band potential of a semiconductor can be estimated by C^−2^ = 0 in Mott-Schottky plots [42]. The TiO_2_ film shows the most positive flat band potential, at −0.21 eV, and the ZnO film shows the most negative flat band potential, at −0.34 eV. Based on the results of UV-Vis absorptance spectra, the band gap of TiO_2_ is 3.2 eV and the band gap of ZnO is 3.10 eV; therefore, the *VB* potential of TiO2 and ZnO is 2.96 eV and 2.76 eV, respectively. The photocathodic protection mechanism of ZnO/TiO_2_ heterojunction film coupled with metal is illustrated in Figure 9b. Under light illumination, electrons in *VB* of TiO_2_ and ZnO in the composite films can be excited to the *CB* of each other. Due to the difference in the position of *CB* and *VB*, as well as the fact that the level of TiO_2_ in *CB* and *VB* is very similar to that of ZnO, photogenerated electrons in the *CB* of ZnO can easily migrate to the *CB* of TiO_2_. Simultaneously, the holes (*h^+^*) in the *VB* of TiO_2_ were transferred to the *VB* of ZnO, and were then captured by the hole scavengers (*HS*, Na_2_S + Na_2_SO_3_) in the electrolyte solution by means of oxidation reactions [43]. The hole scavengers would then be consumed gradually over time, making them an important factor to be solved not only in our system, but also in the photocatalytic system. This band alignment facilitated photogenerated electron-hole separation and migration, lowered the recombination rate, and promoted the quantum yield we need. When irradiated, the photogenerated electrons migrated to the conduction band to be the free electrons. A large number of free electrons will flow to the surface of the metal, causing the metal to have a net electronic surplus state, and the potential of the metal will decrease to a negative potential owing to the polarization. It will be difficult for the metal atoms to lose electrons, and thus they will be protected from corrosion.

## 4. Conclusions

We fabricated ZnO/TiO_2_ double layered heterojunction film with enhanced photocathodic protection efficiency on an FTO substrate using the solvothermal method. Compared with the pure TiO_2_ and pure ZnO film, the composite film ZT1 and ZT2 had higher charge carrier efficiency and lower recombination rate, and exhibited superior photocathodic performance, which can be attributed to the formation of the heterojunction. Compared with ZT1 film, ZT2 film had better performance, because the ZnO nanopraticle layer in ZT2 helps to form the uniform growth of ZnO nanorod arrays, and thus helps to form a better heterojunction electric field at the interface between TiO_2_ nanorods and ZnO nanorods, thus promoting the separation rate of the photogenerated electron-hole pairs. The small difference in band gap and the position of *CB* and *VB* between TiO_2_ and ZnO played an important role in achieving efficient charge separation and promote the photocathodic protection performance.

## Figures and Tables

**Figure 1 materials-12-03856-f001:**
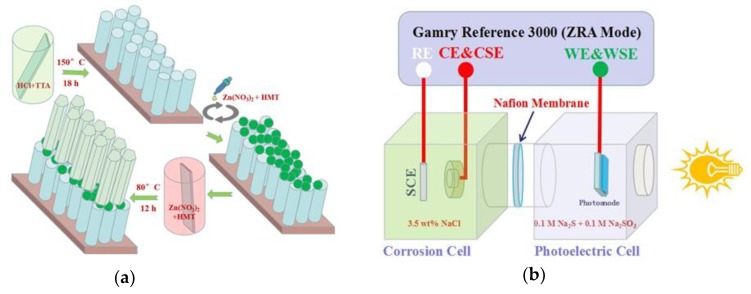
(**a**) Schematic illustrations of the preparation process of the ZnO/TiO_2_ heterojunction double layer film. (**b**) Schematic diagram of the double electrolytic cell setup for the measurement of OCP and photocurrent density.

**Figure 2 materials-12-03856-f002:**
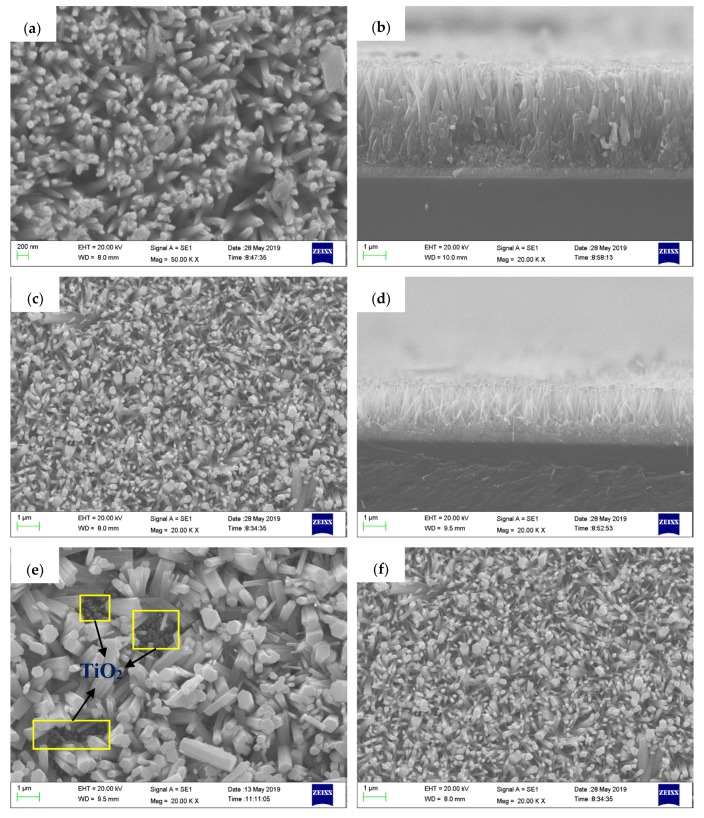
SEM images of (**a**) TiO_2_ nanorod arrays. (**b**) Sectional view of TiO_2_ nanorod arrays. (**c**) ZnO nanorod arrays. (**d**) Sectional view of ZnO nanorod arrays. (**e**) ZT1 film. (**f**) ZT2 film grown on FTO substrate.

**Figure 3 materials-12-03856-f003:**
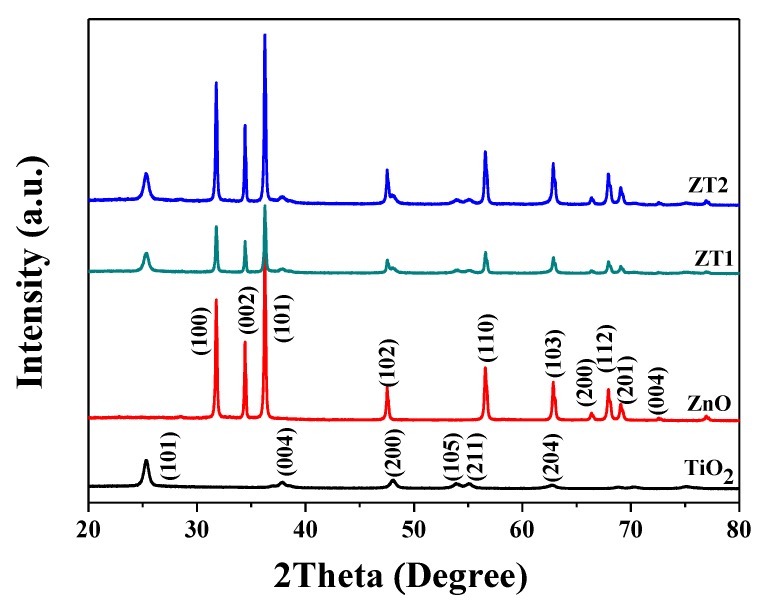
XRD patterns of different photoanode films.

**Figure 4 materials-12-03856-f004:**
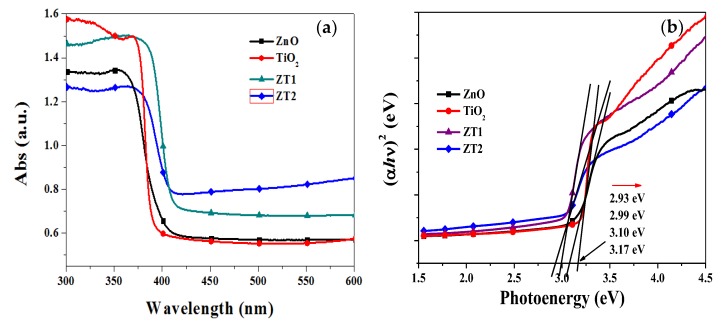
(**a**) UV-Vis absorption spectra, (**b**) plots of (*αhν*)^1/2^ vs. photoenergy of TiO_2_, ZnO, and ZnO/TiO_2_ heterojunction films measured at room temperature.

**Figure 5 materials-12-03856-f005:**
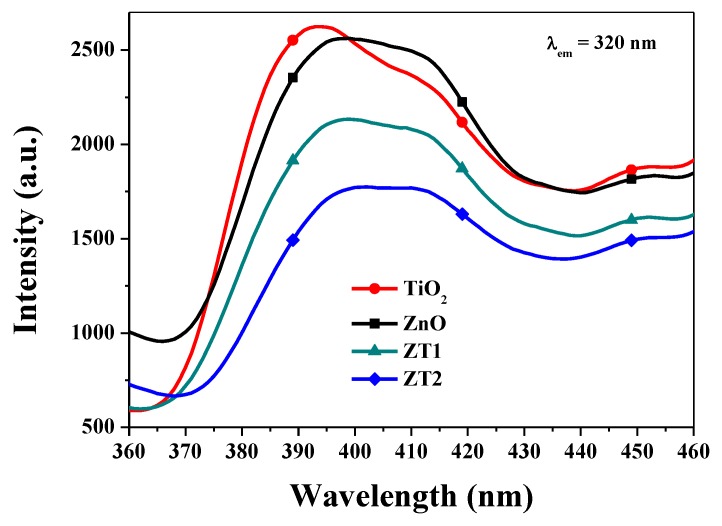
PL spectra of the prepared films (the emission wavelength is 320 nm).

**Figure 6 materials-12-03856-f006:**
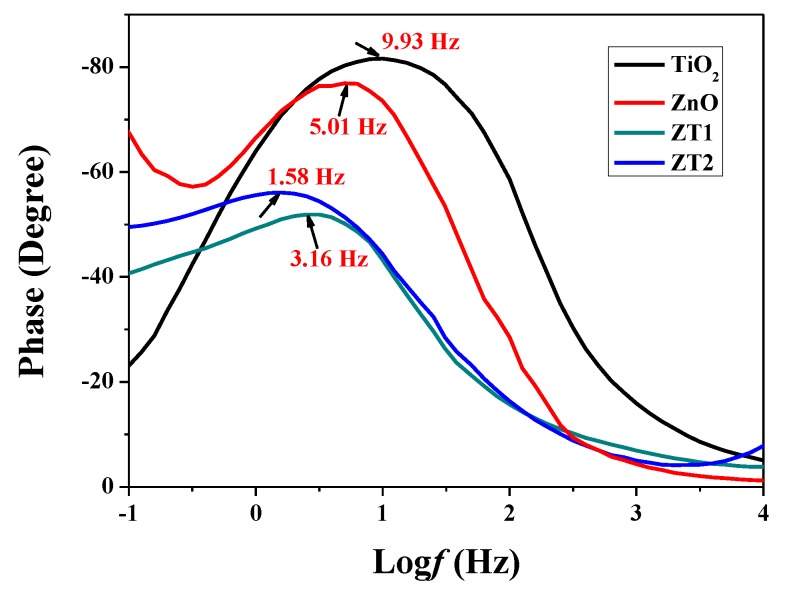
Bode-phase plots of the prepared photoanode films measured in 0.1 M Na_2_SO_4_ solution in dark state.

**Figure 7 materials-12-03856-f007:**
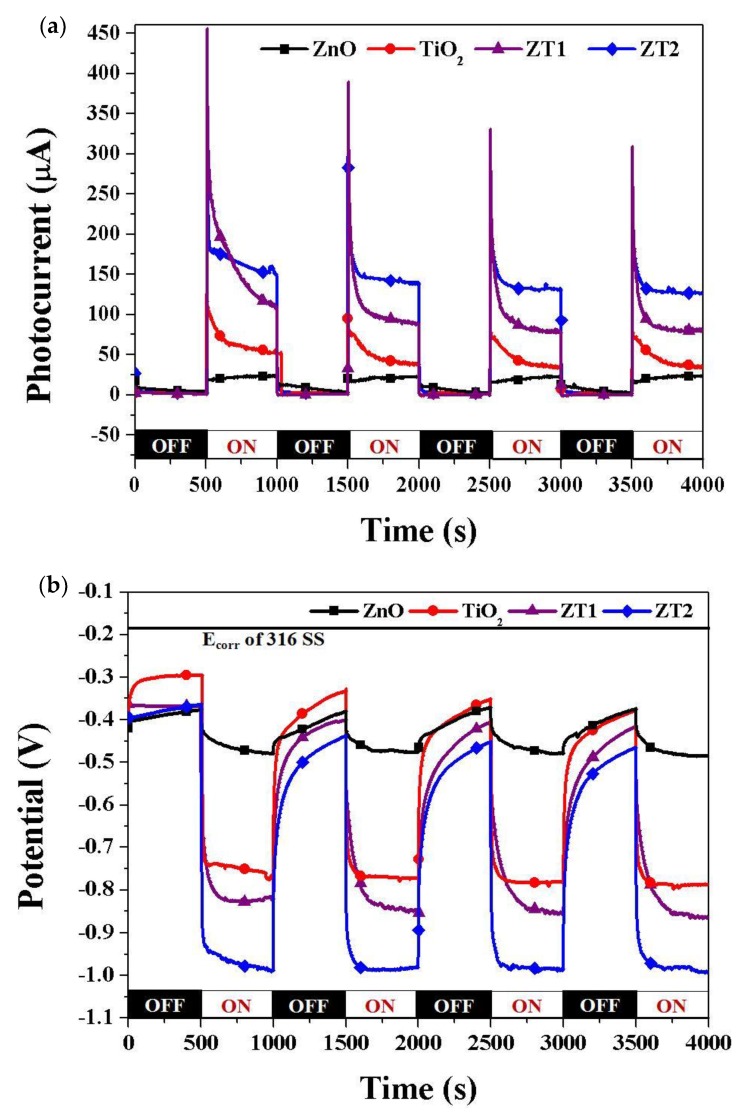
(**a**) Photocurrent density and (**b**) OCP of 316 SS electrode coupled with different films under intermittent simulated sunlight in 3.5 wt. % NaCl solution.

**Figure 8 materials-12-03856-f008:**
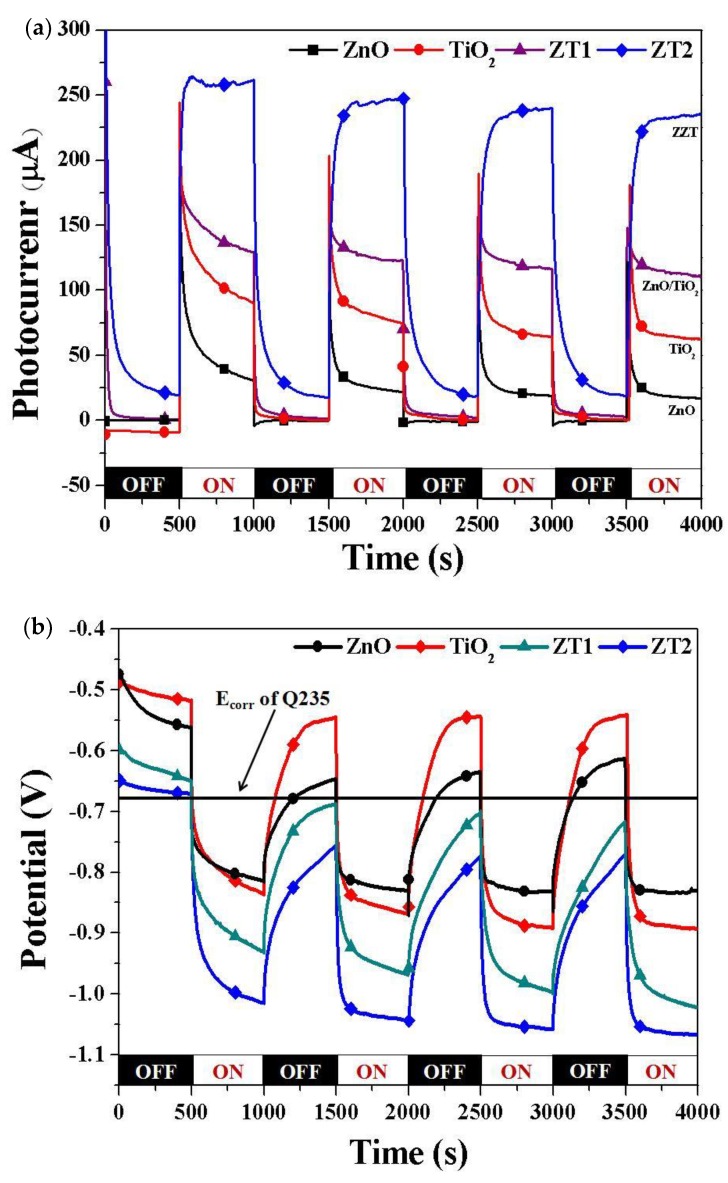
(**a**) Photocurrent density and (**b**) OCP of Q235 electrode coupled with different films under intermittent simulated sunlight in 3.5 wt% NaCl solution.

**Figure 9 materials-12-03856-f009:**
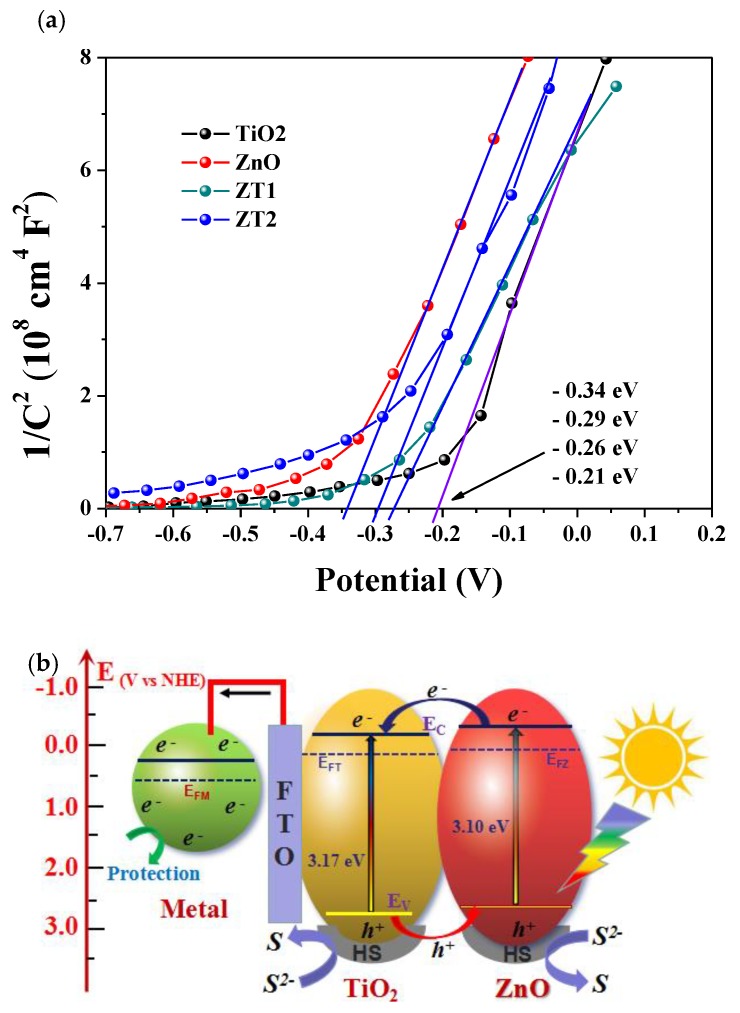
(**a**) Mott-Schottky plots of the photoanode films in 0.1 M Na_2_SO_4_ solution in dark state. (**b**) Schematic illustration of the band alignment and the photogenerated electron-holes migration mechanism in the ZnO/TiO_2_ heterojunction film under simulated solar light illumination for photocathodic protection.

**Table 1 materials-12-03856-t001:** The element content of 316 SS and Q235.

Type	C	Mn	Si	S	P	Ni	Cr	Mo
316SS	0.08	1.80	0.90	0.029	0.045	14.00	17.00	2.00
Q235	0.19	0.59	0.30	0.05	0.44	-	-	-

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
