# Peer review of "Preparation and Photocathodic Protection Properties of ZnO/TiO2 Heterojunction Film Under Simulated Solar Light"

_materials, 2019, doi:10.3390/ma12233856_

Round 1

Reviewer 1 Report

Abstract : ZnO /TiO2 nanorod array redicing corrosion.

OK, no comment.

Introduction :

Clear. References OK. Justification of research and benefits with regards to alternatives such as sacrificial anodes.

Experimental :

Method is clear and succinct, figures are good. The "minor modification" is not spelled out and must be clarified by reading references - this is fine.

(spelling : "after reacted" => "after reacting")

Characterisation section : Correct "wildly used steel" to "widely used steel" (check spellcheckers :).

Results :

Clearly presented - good presentation of structural and compositional and optical absorption / PL data : No comment. The concluding core of the paper on cathodic protection is detailed and convincing.

Conclusions : Succinct and to the point.

Overall comments : This is a quick review because this is good work and there is little to say other than the topic has been well presented and the work described in sufficient but not excessive detail.

Reviewer 2 Report

The authors studied the photocathodic protection properties of a ZnO/TiO2 film supported on FTO. The test were performed in an electrochemical cell with one chamber containing the metal (to be protected) as the cathode and another chamber with the photoanode. The potential for metal protection (aiming for a low corrosion potential) was evaluated based on the electrochemical experiments (photocurrent, OCP). The authors present a systematic study including a detailed characterization of the photoanode (XRD, BGE, PL, …). The main idea is to obtain electrons from the light irradiation of the prepared films that are transferred to the metal. In principle, the results are well presented and conclusions are supported by the experiments. However, some improvements and minor corrections should be done before I can recommend the paper for publication. [1] Some parts of the experiment could be better described, e.g. “First, cut out the prepared photoanode film with a certain size (10 mm × 13 mm, 121 including 10mm × 3 mm blank FTO with no photoanode film). Second, connect the conductive side of FTO with the copper wire by silver conductive adhesive to ensure the good contact. Third, encapsulate the blank FTO and the connected joint with ER.” These parts sound like a instruction rather than a good description of the experiment. [2] The electrolyte is in this case the sacrificial agent to fill the holes after light irradiation. It should be consumed with time and will become a cost factor. Further, what will happen with the electrons at the metal? The authors might explain the process a little but more detailed. [3] Why 3.5 wt% NaCl was used. Of course, it’s the concentration in sea water, but corrosion might appear not only in sea water? [4] Can the authors give the life-time for the charges from the PL experiments? [5] Scheme (b) in Figure 9 might be slightly changed. So it seems that there is a physical contact between Metal and TiO2, which is wrong. [6] The more negative potential might indicate a protection against corrosion, but how can this be quantified? In general, corrosion means loss of metal with time. In the case of protection, the metal loss show be lower. How much lower it will be using the photocathodic protection? Why not performing an experiment by measuring the weight of the metal electrode without and with protection? Thereby, using a stronger corrosive medium. I assume that this information is important as a more negative potential does not mean no corrosion, or? [7] Please specify “The measurements were carried out under simulated solar light source with a 300 W Xenon lamp (Microsolar 300, Beijing Bofeilai Co.).” Was the full lamp spectrum used? If yes, it’s not comparable with solar light source or any other light source, which should be used for protection. What was the light intensity? This also effects the photoanode properties. No one would use a 300 W Xe lamp to protect the metal because of high energy costs. [8] The authors should mention in which area such a protection system should be established (which metal, in which application might be protected) and how it will work sustainable and economic feasible.

Reviewer 3 Report

The present manuscript presents a photocathodic protective structure with ZnO/TiO2 microwire configuration for the carbon steel and stainless steel substrates. The demonstrated approach is not novel, and quite familiar in PEC fields as well as in corrosion field. Unfortunately, no broad impact is observed in the manuscript. In addition, the followings should be carefully resolved:

Page 2, line 55 – The most widely used filed is PEC field, including water splitting and solar redox flow battery applications (RSC Adv., 2018, 8, 6331–6340; Nano Energy, 2016, 24, 10–16; ChemElectroChem, 2019, 6, 106–109). Combination of the TiO2 with narrow band-gap materials is particularly important because the narrow Eg materials tend to easily be dissolved in the electrolyte. Page 4, line 130 – It seems that the used electrolyte has pH around 10.8. This should be mentioned in the manuscript. Besides, the ZnO is quite pH-sensitive material. It is worth to mention that the suggested approach is valid only for high pH environment. Page 5, line 181 - …due to the difference between TiO2 and ZnO… - This part should be rewritten with more detail so that readers can find what characteristics are exactly different. Page 5, line 186 – Fig. 3 - There is also an obvious difference between the ZT1 and ZT2 in XRD, and thus this should be described in the manuscript. Page 5, line 202 – Direct comparison in Abs. is meaningless, since the thicknesses for the samples are not the same. Page 12, line 313 – The described Mott-Schottky analysis is valid only under the assumption that the Fermi-level of the metal oxides is close to the CB. The difference between the Fermi-level and CB should be analysis, and one can apply this to the bend-bending at the interface. This step is very important to understand the charge transfer or tunneling across the interface of this kind of junctions ( Chem. Educ., 1983, 60, 327).

Reviewer 4 Report

Manuscript entitled "Preparation and photocathodic protection properties
of ZnO/TiO2 heterojunction film under simulated  solar light" by Xiong Zhan and co authors described a  ZnO and TiO2 nanorod arrays prepared on FTO glass by a solvothermal method and their study towards photocathodic performance. Presented research are well design and have a suficient scientific level. However, manuscript have to be improved prior to its acceptance. 

My detailed comments:

1) the novelty of research is not discussed in a proper way, please highlight a novel aspect of the material

2) manuscript seems to be a "list of result" without real scientific discussion. Authors should to extend a discusion and conclusion.

3) Authors review an examples of composites of TiO2 with other semiconductor. It is important to add to the introduction 1 sentence about composites with zinc oxide. Please read and cite recent literature e.g.  International Journal of Hydrogen Energy 2019, 44 (50), pp. 27343-27353 

4) In introduction authors should mention about other examples of heterojuntions e.g. including copper compounds (Electrochimica Acta
2018, 266, pp. 441-451)

5) It would be interesting to see a differences in  Photocurrent density in diffrent pH e.g. pH 3 and 9.

Round 2

Reviewer 2 Report

The authors have not respond to the following points from earlier submission so that revision is necessary. [1] The electrolyte is in this case the sacrificial agent to fill the holes after light irradiation. It should be consumed with time and will become a cost factor. Further, what will happen with the electrons at the metal? The authors might explain the process a little but more detailed. [2] Why 3.5 wt% NaCl was used. Of course, it’s the concentration in sea water, but corrosion might appear not only in sea water? [3] Can the authors give the life-time for the charges from the PL experiments? [4] Please specify “The measurements were carried out under simulated solar light source with a 300 W Xenon lamp (Microsolar 300, Beijing Bofeilai Co.).” Was the full lamp spectrum used? If yes, it’s not comparable with solar light source or any other light source, which should be used for protection. What was the light intensity? This also effects the photoanode properties. No one would use a 300 W Xe lamp to protect the metal, because of high energy costs. [5] The authors should mention in which area such a protection system should be established (which metal, in which application might be protected) and how it will work sustainable and economic feasible.

Reviewer 3 Report

I concede that the manuscript has been improved significantly after the authors have addressed the reviewers' comments thoroughly. Most of the responses from the authors are acceptable and understandable.
